# Managing Quartz Exposure in Apartment Building and Infrastructure Construction Work Tasks

**DOI:** 10.3390/ijerph20085431

**Published:** 2023-04-07

**Authors:** Tapani Tuomi, Tom Johnsson, Arto Heino, Anniina Lainejoki, Kari Salmi, Mikko Poikkimäki, Tomi Kanerva, Arto Säämänen, Tuula Räsänen

**Affiliations:** 1Finnish Institute of Occupational Health (Työterveyslaitos), P.O. Box 40, FI-00032 Helsinki, Finland; 2Tapaturva Ltd., Finnoonlaaksontie 2, FI-02270 Espoo, Finland; 3Lotus Demolition Ltd., Kalliosolantie 2, FI-01740 Vantaa, Finland

**Keywords:** respirable crystalline silica, respirable dust exposure, exposure management, construction workers, silicosis

## Abstract

The present report describes exposure to respirable silica and dust in the construction industry, as well as means to manage them. The average exposure in studied work tasks (*n* = 148) amounted to 64% of the Finnish OEL value of 0.05 mg/m^3^. While 10% of exposure estimates exceeded the OEL, the 60% percentile was well below 10% of the OEL, as was the median exposure. In other words, exposure was low in more than half of the tasks. Work tasks where exposure was low included construction cleaning, work management, installation of concrete elements, rebar laying, driving work machines equipped with cabin air intake filtration, and landscaping, in addition to some road construction tasks. Excessive exposure (>OEL) was related to not using respiratory protection at all or not using it for long enough after the dusty activity ceased. Excessive exposures were found in sandblasting, dismantling facade elements, diamond drilling, drilling hollow-core slabs, drilling with a drilling rig, priming of explosives, tiling, use of cabinless earthmoving machines, and jackhammering, regardless of whether the hammering took place in an underpressurized compartment or not. Even in these tasks, it was possible to perform the work safely, following good dust prevention measures and, when necessary, using respiratory protection suitable for the job. Furthermore, in all tasks with generally low exposure, one could be significantly exposed through the general air or by making poor choices in terms of dust control.

## 1. Introduction

Crystalline silica has seven different crystal forms. In Finland, you can be exposed mainly to quartz and, in the ceramics industry or in the production of refractory materials, also to cristobalite [1]. Quartz occurs in bedrock minerals, sand, and rock materials. An estimated 12% of the Earth’s crust consists of quartz [2]. Therefore, quartz is also common in products and materials containing minerals, sand, and rubble. In construction, such materials include concrete, cement, mortar, bricks, tiles, sand, stones, and bedrock. The physical properties of crystalline silicas and, hence, their insolubility in the lungs make them harmful, potentially causing silicosis and cancer upon long-term exposure to respirable particles. Since the lung changes affected by silicosis, as well as lung cancer, are irreversible and there is no effective treatment for them. Thus, all efforts must be made to prevent or reduce exposure of workers to quartz dust [3].

The health hazards associated with stone dust exposure are widely recognized, and their causes, consequences, and means of prevention have been known for a long time. Even so, quartz exposure is still one of the biggest causes of health harm and increased risk of illness in many professions. In Finland alone, at least 50,000 workers are exposed to quartz in their work, with 0.2‰ of them per year officially contracting either silicosis or lung cancer derived from quartz exposure at work [4]. In order to promote the well-being of those exposed to quartz at work and to reduce the economic and human losses resulting from exposure, industrial sectors where quartz exposure is a problem in the European Union (EU), excluding the construction industry, signed “The Agreement on Workers Health Protection through the Good Handling and Use of Crystalline Silica and Products Containing it” in 2006 (NEPSI treaty) [5]. Both the employer and employee organizations were parties to the agreement. With the agreement, occupational hygiene limit values and exposure assessment methods for respirable quartz were harmonized in some EU countries, including Finland [6].

In the first five years after the NEPSI treaty, work-related exposure to respirable quartz in our country fell sharply in the sectors that had joined the agreement. Nowadays, the level of exposure has stabilized, and exposure exceeding the OEL are significantly rarer than before [4,7]. Consequently, the 95 percentile of respirable quartz workplace exposure measurements made by the Finnish Institute of Occupational Health (FIOH) was 1.0 mg/m^3^ in 2006 and 0.05 mg/m^3^ in 2016, while the share of exposures exceeding the OEL value decreased simultaneously from 44 percent to 5 percent [4]. However, the reduction in work-related exposure would seem to apply mainly to industry sectors that signed the NEPSI treaty, but not as much to construction work. Construction work accounted for only 6% of the occupational hygiene exposure measurements made by FIOH until 2020, despite 2/3 of those exposed to quartz at work being construction workers [4,7].

### 1.1. Statutory Employer Obligations in Finland Relevant to Respirable Silica Exposure

According to the Finnish Law on safety at work, the evaluation of risks pertaining to work tasks is obligatory, and hence, so are exposure assessments of respirable quartz [8]. Unless excessive exposure can be ruled out altogether or risk assessment can be based on available data, this necessitates exposure measurements. The need for occupational hygiene measurements and, with that, the investigation of exposure to respirable silica is also included in the government decree on the safety of construction work and on the list and register of those exposed to carcinogenic substances and methods at work (ASA register) [9,10]. In addition, the government decree on the safety of construction work stipulates that on a joint construction site, both the developer’s instructions and the main contractor’s safety plans must give directions on the procedures for occupational hygiene measurements in the construction project.

At the end of 2019, a binding limit value based on cancer risk (2xOEL value, i.e., 0.1 mg/m^3^) was set for respirable silica in Finland in the government decree on the prevention of work-related cancer risk that must correspond with the value set in EU directive 2019/130 [11,12]. With this, the act on the list and register of those exposed to carcinogenic substances and methods at work (ASA register) was revised [10]. As a result, those exposed to respirable quartz at work must be reported to the ASA register. In practice, it took these two measures for the construction sector to finally comprehensively recognize that the exposure of construction workers to quartz needs to be investigated and managed.

According to current practice, workers who are exposed to carcinogenic factors more than the background concentration are reported to the ASA register. If their quartz exposure exceeds what they would be exposed to without the job in question, registration is obligatory. In Finland, concentrations of respirable quartz in general air have not been published, and there is very little information on background concentrations in the rest of Europe. More measurements have been made in the USA. The concentration of respirable quartz in US metropolises ranges from 0.0011 to 0.0088 mg/m^3^, with an average of 0.0032 mg/m^3^ [13]. In measurements made in 1991 in the vicinity of a sand quarry in Monterey, California, the concentration of respirable quartz was 0.0011–0.0013 mg/m^3^ [14]. In one similar measurement made by FIOH, the concentration was 0.00020–0.00023 mg/m^3^ (unpublished data). According to the US Environmental Protection Agency (EPA), 0.0050 mg/m^3^ is a low enough concentration to protect citizens from the harmful effects of respirable quartz [15]. However, some US states have stricter health-based maximum concentrations for respirable quartz in ambient air. For example, in Texas, it is 0.00027 mg/m^3^ for long-term exposure, and in California and Vermont, 0.0030 mg/m^3^ and 0.00012 mg/m^3^, respectively [16]. In Finland, the practice is that employees are reported to the ASA register if they are estimated to be exposed to more than 0.0050 mg/m^3^ of respirable quartz at work, i.e., 10% of the present OEL value. Based on the above, this is a level that is rarely exceeded, even in urban environments. Furthermore, this is also a level that, according to current knowledge, is not associated with mentionable health hazards because human lungs are able to remove small concentrations of quartz to the ciliary area, from which they are removed via the mucus elevator [17].

In addition to ASA registration, according to Finnish legislation, employers must organize health checks for workers in work tasks that are associated with special risks of illness [18]. In the case of respirable quartz, this means that regular monitoring of the state of health must be arranged for workers who are exposed to the extent that, according to current knowledge, causes a specific risk of illness. Meaning, in practice, that quartz exposure is significant if it is greater than 0.02 mg/m^3^ as a working career average [19,20,21,22].

### 1.2. Exposure Classification

As mentioned, the (on principle) health-based OEL for quartz and other respirable silicon dioxides set by the Finnish Ministry of Social Affairs and Health is 0.05 mg/m^3^. Finnish OEL values are set for air pollutants in the workplace, which “the employer must take into account in the investigation and assessment of hazards and in the planning of the work environment when evaluating the cleanliness of the workplace air, and the exposure of employees” [6]. The EU’s so-called limit value set in the “directive on the protection of workers from the risks related to exposure to carcinogens or mutagens at work,” and ratified in the Finnish Government Regulation 1267/2019 is 0.1 mg/m^3^ [6]. This value is not health-based, but it is a statutory limit that should not be exceeded in terms of average exposure over the course of a working day. The OEL values are basically health-based, but when setting them, the seriousness of the exposure-related health damage is considered, as is the level that is technically and economically possible in the workplace with current technology [6].

In the present study, a FIOAH exposure classification was adapted to respiratory exposure to quartz and respirable dust, with the difference that the limit for significant quartz exposure was lowered from 50% to 40% of the OEL (Table 1) [23]. The basis being the EU risk assessment committee’s estimation that <40% of the OEL, as a working career average, and which corresponds to a low risk level for silicosis [19]. Furthermore, most risk assessments on careerlong-quartz-exposure-associated risks, a concentration corresponding to the OEL, are associated with an increased silicosis risk [20]. Analogously, cumulative lung cancer risk estimates often end up between 0.04–0.05 mg/m^3^ for a level of low cancer risk [21,22].

In the case of respirable dust, the results were compared with the OEL value of respirable cement dust (1 mg/m^3^) because the effects of chromates and quartz are not taken into account in its basis. Furthermore, many—although not all—of the dusts we measured on construction sites are quite comparable to cement dust in their composition, alkalinity, and physiological effects [24,25]. This is a higher concentration than the FIOH target level for general respirable dust (0.5 mg/m^3^) but lower than the corresponding OELs in some EU countries (2.5–5 mg/m^3^) [26,27,28].

### 1.3. Control of General Air Respirable Dust Concentrations

Indoor construction work is often done in poorly ventilated premises, which is why a high respirable quartz content in the general air may, in many professions, be a major source of exposure, including work tasks that do not generate quartz dust per se. According to the Finnish Government Regulation on the safety of construction work, dust must be removed via air conditioning, targeted removal, or other appropriate measures [11]. If necessary, the spread of dust must be prevented by shielding dusty work areas through protective walls, with the shielded area underpressurized to yield a sufficient pressure difference to inhibit the spread of dust, as well as an air exchange ratio of, at the minimum, 6 h^−1^ [29]. Compartmentalization and negative pressurization are, however, most often not possible in conventional work due to economic constraints. Hence, our partners and the representatives of the construction sites we investigated did not consider negative pressure to be a realistic option in most conventional interior work tasks. According to their estimation, it would substantially increase work and heating costs. 

Air recirculating cleaners equipped with HEPA H13 filters may be a good way to supplement dust control, based on experience gained from Swedish construction sites, particularly when they are used in confined dusty departments, together with machine-specific exhaust vents, and possibly also with water to control dust emissions. On the condition, however, that they are placed as close as possible to the dust source [30,31,32,33]. Consequently, in this project, we tested the effectiveness of air-recirculating mobile air cleaners to control dust concentrations of general air in dusty, compartmentalized work areas. An alternative in many jobs would be to use water to bind dust and prevent emissions, but it is often not an accepted option indoors. Nonetheless, compartmentalization coupled with underpressurization remains the only viable option in many renovation work tasks where structures are demolished, as well as other work tasks where dust production is high [29].

A more effective means to control dust concentrations in general air is to limit emissions at their source before they spread to the surrounding air space. Hence, we also wanted to test means of dust control in two critical work-related tasks with respect to dust emissions to the general air, the first being the mixing of mortar indoors and the second being the use of drill rigs outdoors.

### 1.4. Aims of the Study

Most domestic construction companies have not estimated the quartz exposure of their workers as part of statutory risk assessment. Hence, construction employers often do not have sufficient data to assess who they should report to in the ASA register and in which tasks the statutory limit value or OEL value may be exceeded [4]. Furthermore, that is also why it is challenging for many, if not most, construction companies to decide whose employee’s health should be monitored by the occupational health care contractor. These are considerable practical problems in an industry where tens of thousands of workers are potentially exposed to quartz. This study addresses these needs. The starting point was to identify work tasks where respirable quartz exposure can be moderate, significant, or excessive. Furthermore, we aim to give instructions on their safe execution so that the exposure can be kept low (Table 1). To serve this end, the project also evaluated the effectiveness of various dust control measures in reducing exposure. Finally, the goal was to list tasks where occupational health care, health monitoring, and/or ASA registration come into question.

This report is an extension of a Finnish research report that included additional data not contained here [34]. The bulk of the report is available online in English [35]. Exclusions include good practices for 29 work tasks and four general procedures, in addition to estimates of work tasks, method-specific exposure ranges, and more detailed information on the applicability of direct-reading measurements and test results pertaining to the management of quartz emissions in selected work tasks [34,35,36].

## 2. Materials and Methods

### 2.1. Selection of Work Tasks to Be Measured

A survey aimed at the project’s participants was used in the selection of construction work tasks associated with respirable quartz exposure. In addition to the executors (FIOH, Tapaturva Ltd. (Espoo. Finland), and Lotus Demolition Ltd. (Vantaa, Finland)), the participants were the Confederation of Finnish Construction Industries RT, the Finnish Construction Trade Union, the Regional State Administrative Agency (workplace inspectorate), and the five companies mentioned below (Section 2.2). Based on the survey, a list of work tasks to be investigated in different phases of new and renovation construction was drawn up. The phases included foundation and frame stages of new construction, yard work, interior work phases in the renovation and novel construction of apartment buildings, facade work, demolition of buildings, infrastructure construction, concrete waste, and stone crushing (pulverization). Selections were refined as the project progressed while planning measurement visits in cooperation with the construction sites and were based on the information gathered at the construction sites.

### 2.2. Sampling Sites

A total of 58 construction sites were visited on 63 different days between 1 January 2021 and 31 September 2022. The exposure to respirable quartz and dust during work days and during dusty chores was measured separately for a total of 148 workers, yielding 296 samples for the analysis of respirable dust and as many for the analysis of respirable quartz. The general air concentrations of respirable dust and quartz were estimated with the help of 88 stationary samples, with a minimum of one sample from each construction site included. The measurements were fairly evenly distributed amongst the five companies that participated in the project from the beginning, which were NCC Finland Ltd. (Helsinki, Finland) and Hartela Ltd. (Turku, Finland) (construction of new apartment buildings), CONSTI Ltd. (Helsinki, Finland) (renovation of residential and public buildings as well as office buildings), Destia Ltd. (Helsinki, Finland) (road and infrastructure construction, stone breaking plants), and Lotus Demolition Ltd. (Vantaa, Finland) (demolition of buildings). Moreover, two additional companies were recruited to provide exposure measurement sites for the study, Purkupiha Ltd. (Helsinki, Finland) and Mevaset Ltd. (Ylöjärvi, Finland) (mobile concrete crushing plants). 

In addition to exposure measurements of dust-generating work tasks performed on actual building sites, the effectiveness of dust control measures was measured when drilling with drilling rigs, in the use of air-recirculating mobile air cleaners, and in mortar mixing points. Furthermore, task-specific exposure was monitored while following different types of dust and exposure prevention measures, including compartmentalization and underpressurization, machine-specific exhaust vents and irrigation, and respiratory protection.

### 2.3. Measuring Strategy

Many of the dust-producing tasks in construction require the use of a respirator for part of and, occasionally, the entire workday. In each workplace we studied, our goal was to find out both the average exposure of the employees throughout the working day and the exposure concentration during dusty work tasks requiring respirators. For this reason, two cyclones collecting respirable dust were hung in the breathing zone of the workers. The air sampling pump of the first sampler was on for the entire work shift, yielding the average exposure concentration throughout the workday. In the second sampler, the air was only sampled when the worker was wearing a respirator. If a respirator was not used by the worker, the second sampler was started whenever, according to the researcher’s assessment, a respirator should have been used. By taking into account the assigned protection factors of the respirators used, we were able to collect the following exposure-describing results from the workers:Exposure concentration during the working day (from the first sampler)Exposure concentration in dusty chores (from the second sampler)Average exposure throughout the working day (see Equation (1) below)

In the case of workers who did not use respirators:4.Average exposure throughout the working day if a respirator suitable for the job had been worn during dusty activities.
(1)$$E_{wd,2}={(C}_{wd}\times t_{wd}-C_{wp}\times t_{wp}\times {(PF}_{a}-1)/{PF}_{a})/t_{wp}$$
where,
$C_{wd}$ = Average concentration during the working day in the breathing zone of workers (mg/m^3^)$C_{wp}$ = Average concentration during dusty undertakings in the breathing zone of workers (mg/m^3^) = exposure of unprotected workers in dusty chores.$t_{wd}$ = sampling time of working day sample.$t_{wp}$ = sampling time of sample withdrawn during dusty chores.${PF}_{a}$ = Assigned protection factor of a respirator: 20 for FFP3 masks, 500 for face covering masks with P3 filters, 200 for P3 masks with assisted breathing, and 1000 for when ex air pressured sand-blowers phase masks were used.$E_{wd,1}$ = $C_{wd}$ = Average exposure of workers during the working day if he or she did not use a respirator.$E_{wd,2}$ = Average exposure of workers during the working day, taking into account the respirator that was used.$E_{wd,3}$ = $E_{wd,2}$= Estimate of exposure of workers working without respiratory protection if he or she had worn a respirator during dusty activities.

In addition to employee-specific samples describing the actual exposure, samples describing the general air concentration were also collected from stationary measurement points at each workplace. These samples were collected from the same floor, department, or apartment where dust-emitting work was carried out. In the case of outdoor work, general air samples were taken from the cabins of work machines and other places where workers were exposed to respirable quartz through the general air. Such places included blasting sites, earthmoving sites, green buildings, road construction sites, and rail sites.

### 2.4. Sampling and Analysis of Samples

The sampling of respirable dust was performed as described in CEN and ISO standards [37,38]. Briefly, airborne respirable dust samples were collected on 25 mm mixed cellulose ester membrane filters (Millipore AAWP025, 0.8 µm, Merck KGaA, Darmstadt, Germany) using SKC GS-3 nylon cyclones (SKC Inc., Philadelphia, PA, USA). The flow rates of sampling pumps were calibrated with an accuracy of ±5% to comply with the respirable fraction: 2.75 dm^3^/min. Samples were collected either from the breathing zone of workers or from stationary points at a height of approximately 1.5 m. Sampling was continued for a minimum of 4 h, usually close to 8 h, to estimate the average 8 h exposure of workers.

Respirable dust was analyzed gravimetrically as described in ISO 15767 [39]. Briefly, samples were first dried in desiccators for 2 days prior to conditioning in standard conditions (temperature 20 ± 2 °C, relative humidity 50 ± 5%) for a minimum of 24 h. Conditioned filters were weighed using a Mettler Toledo XP56 precision balance (Mettler-Toledo AG, Greifensee, Switzerland) with a readability of 0.001 mg. The limit of quantitation (LOQ) of the gravimetric analysis was 0.06 mg. In total, 31% of the respirable dust analyses in the present study fell below the LOQ. The β-substitution method introduced by Ganser and Hewett [40] was used to calculate a β_GM_-factor for adjusting each value below LOQ. The β_GM_ calculated from uncensored respirable dust data was 0.53, and the values below LOQ were therefore substituted with 0.53 × 0.06 mg (0.03 mg).

Samples containing calcite, i.e., most of the samples withdrawn, were treated with HCl of analytical grade (VWR Chemicals, Paris, France) prior to quartz analysis. Namely, the mixed cellulose ester filters containing the sampled dust were placed in a filtering funnel (pore size 0.5 µm, diameter 25 mm) using tweezers. A combination of 10 cm^3^ of 9% HCl and 5 cm^3^ of 2-propanol of analytical grade (VWR Chemicals, Paris, France) was added, and the sample filtered with the help of a vacuum pump after 5 min. The filter was washed twice with 15 cm^3^ of deionized water and left to dry overnight in porcelain crucibles using tweezers. Crucibles containing the dried samples were covered and ashed (2 h, 600 °C). Ca. 300 mg of oven-dried (110 °C, 24 h, stored in a desiccator) and mortar ground KBr of infrared quality (Merck kGaA, Darmstadt, Germany) was added to the crucibles, after which the sample was transferred to a mortar using a wooden spoon and a camel hair brush. The samples were ground with the help of a pestle under a heat-generating lamp. Lastly, the sample was transferred to a pellet-pressing platform, and the pellets were pressed with standard technology using a Specac pellet press (Specac Ltd., Orpington, UK). Blank samples and control standards were prepared in an identical way.

Samples and standards were measured as described in NIOSH method 7602 [41]. The IR spectra were measured in absorbance mode. The pellet was scanned from 1000 cm^−1^ to 600 cm^−1^, and the peaks 775 and 800 cm^−1^ were used to identify quartz. Quantification was based on the absorbance (peak height) at 800 cm^−1^, using the mean of four consecutive measurements. If the variation of the four measurements exceeded three times the standard variation from 60 random validation samples, the KBr tablet was reground, repressed, and remeasured. The LOQ of the quartz analysis was 2.0 µg/sample. Of the respirable quartz analyses in the present study, 21% fell below the LOQ. As with the respirable dust analyses, the β-substitution method introduced by Ganser and Hewett [40] was used to calculate a β_GM_-factor for adjusting each value below LOQ. The β_GM_ calculated from uncensored respirable quartz data was 0.56, and the values below LOQ were therefore substituted with 0.56 × 2 µg (1 µg). If the result exceeded the linear range of the analysis (200 µg/sample), the sample was reground and diluted with KBr before remeasurement.

### 2.5. Effectiveness of Selected Dust Control Measures

Dustcontrol DC AirCube 2000 mobile air cleaners (Dustcontrol Ltd., Norsborg, Sweden), equipped with HEPA H13 filters and with an airflow rate of 1800 m^3^/h, were used in intervention measurements where the same indoor work was performed during day one without using the cleaners and on day two with the cleaners. In the tests, the air cleaners were placed within 1–3 m of the dust source, with the air inflow directed towards the source (Figure 1).

In terms of dust control, the most effective way is to remove the generated dust at its source. Therefore, we also wanted to test the effectiveness of controlling dust emissions in the mixing of mortar with a mobile air cleaner. The cleaner had an airflow of 1300 m^3^/h and was equipped with airflow-directing side screens and an M-class filter (IEC 60335-2-69, Camu D2, Consair, Helsinki, Finland, Figure 2). These measurements were done at an actual indoor work site. As was the case in all other indoor worksites visited, there was no general ventilation on the premises.

For the same purpose, we tested an exhaust vent (Bad-Dust Oy, Helsinki, Finland) attached to the mixer and connected to a Dustcontrol DC-TROMB 400 vacuum cleaner with an airflow of 300 m^3^/h as measured from the exhaust vent. The vacuum cleaner was equipped with a HEPA H13 filter. These measurements were done in a test chamber with a ventilation rate of 0.7 h^−1^ and a surface area of 27 m^2^, using a Flex R 503 FR, 530 r/min blender with a blade length of 680 mm (Steinheim, Germany, Figure 3).

The drilling of shot holes and grooves at blasting sites can be associated with excessive exposure of the driller as well as other workers working near the drilling rig. As we did not come across effective enough dust control measures on actual working sites, we decided to test dust prevention measures related to drilling in field tests using a Sandvik Mining DINO DC420Ri top impact drilling rig equipped with 40 L water container (Sandvik AB, Sandviken, Sweden). The water supply in wet drilling was adjusted to 0.4 dm^3^/min. The diameter of the drilling holes was approx. 60 mm. The average concentration of respirable dust and quartz was measured at stationary measurement points and in the driller’s breathing zone while drilling boreholes for approximately 3 h. Measurements were performed (1) without using water and without attaching flexible plastic tubes (socs) to dust separators (cyclones); (2) without water use, but with socs attached to front and rear cyclones directing dust emissions to the ground, thereby minimizing release of dust to the air; and (3) with socs attached to cyclones and with water supply (Figure 4 and Figure 5).

## 3. Results

### 3.1. Work-Task-Specific Exposure

In the work tasks that were investigated, the exposure to respirable quartz was, on average, 0.032 mg/m^3^. The proportion of exposures exceeding 10% of the OEL (ASA registering limit) was 38%, while 10% of exposures exceeded the OEL altogether (Table 2). However, in more than half of the tasks, the exposure was negligible to low, as the 60% percentile was below the ASA registering limit, as was the median exposure (Table 2). Work-task-specific exposures to respirable quartz, taking into account the use of respirators, are shown in Figure 6. The general characteristics of exposures are presented in Table 2.

In terms of exposure, other dust-producing tasks performed nearby or on the same department or floor were decisive because concentrations in the general air exceeded exposure in many work tasks (Table 2 and Table 3). A significant proportion (22%) of the general air concentrations were of a magnitude associated with an increased silicosis risk (>0.2 mg/m^3^, >40% of OEL) when present during the span of a working career (Table 3, Figure 7. High concentrations in the general air requiring respiratory protection were measured in underpressurized departments, where partition walls or suspended ceilings were demolished by a robotic jackhammer, in the pulverization of concrete demolition waste, in gravel pits near crushers, in the loading and spreading of railroad ballasts, in blasting sites near drilling rigs, in hooded facade work, and in departments where jackhammering, hollow-core slab drilling, or diamond sawing and diamond drilling were carried out (Figure 6). On average, the general air concentrations of respirable quartz and dust on the construction sites were higher than the workers’ exposures, both when comparing average exposures and 95 percentiles, as well as percentages exceeding the OELs and the quartz statutory exposure limit (Table 2 and Table 3). Hence, during indoor work in departments where very dusty tasks were performed, all those visiting the department for even a short time should have worn at least an FFP3 respirator. Furthermore, those who worked there for longer periods should have worn a breathable, filterable TH3P/TM3P class respirator. Unfortunately, for the most part, this did not happen, and respirators were usually worn only by those performing the dusty tasks. This was also the case in underpressurized compartments. Such very dusty tasks included, for instance, jackhammering, drilling of hollow-core slabs to remove entrapped water, diamond sawing and diamond drilling, floor grinding, and the leveling of inner walls or roofs (Figure 6).

Excessive quartz exposures, i.e., exposures exceeding the OEL, were measured in all phases of construction (Table 4). Namely during sandblasting, diamond sawing and drilling, demolition hammering, and drilling of hollow core slabs and charging holes (Table 4). Excessive exposures were mostly related to not using respiratory protection at all, not using respirators for long enough after ceasing a dusty activity, or not protecting oneself against high quartz concentrations in the general air (Table 4). Just as the excessive exposures mentioned, significant exposures (>0.2 mg/m^3^) present during the span of a working career are associated with an increased silicosis risk. Significant exposure was found in the leveling of indoor walls and roofs, the spreading of railroad ballasts, and during road construction (footman) (Figure 6).

Based on our measurements, work tasks that expose you to low or moderate quartz concentrations include work management, element installation, rebar placing, driving work machines equipped with cabin air intake filtration, landscaping, and construction cleaning, as well as several tasks related to road construction (Figure 6). However, in all these tasks, you could be exposed significantly, and sometimes even excessively, by making suboptimal choices (Figure 6).

### 3.2. Respirator Use and Exposure

The construction industry work tasks where quartz dust exposure was thought to be most probable were pre-selected for the present measurements. In many of the studied tasks, respirator use was necessary for dusty activities and sometimes throughout the working day. However, as respirators were usually not used for long enough periods outside of dusty activities, they reduced the respirable quartz exposure, on average, by no more than 50% (median 58%). In other words, with the use of respirators, on average, the worker’s quartz exposure was half of what was the average working day concentration in their breathing zone (Figure 8).

The distribution of quartz exposure by concentration category among workers who used respirators did not differ significantly from that of workers who did not use respiratory protection. The proportion of those who exceeded the ASA notification threshold, as well as the proportion of those who exceeded the OEL value or the binding limit value, was surprisingly even higher among workers who used respirators (Table 5). This is largely explained by the fact that, based on our observations, respirators were mostly used when actively doing dusty activities. As a result, after dusty work chores, particularly indoor work, workers were exposed through the general air. Due to poor ventilation, the concentration of dust produced decreased fairly slowly upon ceasing the dusty activities (Table 3 and Table 4).

## 4. Discussion

Since mainly construction work tasks that were seen as challenging in terms of respirable quartz exposure were assessed, the average or median results cannot be generalized to all building or infrastructure work, only the part of the work tasks that were studied. Tasks from all phases of apartment building construction were assessed. This included carpenters, painters, electricians, foremen, and other workers that, during a typical workday, are exposed to quartz through the general air and also during short and far-between activities, such as drilling short holes in concrete or leveling rough patches in walls. Similarly, in infrastructure, we did investigate green builders and road construction workers that, for much of the working day, are not exposed at all. It can therefore be argued that the present study does give a comprehensive view of the exposure of those Finnish construction workers that can be exposed to respirable quartz at work. 

Among these workers, the 60% percentile of estimated exposures was less than 10% of the OEL, which is low and below the ASA registering limit. The 90th percentile of all estimated exposures was 0.05 mg/m^3^, matching the OEL, and the 95th percentile was 0.2 mg/m^3^, exceeding the statutory exposure limit by 100%. Estimates of work-task-specific exposure ranges were highly dependent on the adapted dust prevention measures, including ventilation, water use, and the use of tool-specific exhaust vents, as well as on the use of respirators and the duration of respirator use. In addition, the timing of work tasks with respect to the dust emission potential of simultaneously performed tasks was key in preventing exposure. On average, quartz present in the general air contributed significantly to the workers’ total exposure. Furthermore, this should be taken into account when assessing the need for using respirators outside of actively performing dusty activities. In fact, in many cases, the general air was the main source—and could also be the only source—of quartz dust exposure. Overall, in this project, respirable quartz concentrations in the general air were of a similar level to those previously reported by Antonsson and Sahlberg [33], while the corresponding respirable dust concentrations were somewhat higher (Figure 8). Low general air respirable quartz concentrations (<0.005 mg/m^3^) were mostly measured during outdoor work and in all the break rooms we examined. However, high general air concentrations were also measured outdoors, for instance, at blasting sites and near the crusher at gravel pits, and when pulverizing demolition concrete.

As it stood in the investigated construction sites, the use of respirators, on average, lowered exposures only to half of what they would have been had they not been used at all. To give a typical example, if a worker wearing a respirator with assisted breathing actively grinds concrete floors for 60% of a typical 8 h work day and for 30% of the time, stays on the same premises without the respirator, finishing corners, pipe inlets, and radiator surroundings, and if the average quartz concentration in the premises corresponds to the average general air concentration in our study (144% of OEL), his average exposure during said work day will amount to 43% of the OEL. This means that even though he wears a respirator during active grinding, the risk that he will contract silicosis may still be elevated, provided the day in question is a good measure of his average career exposure to respirable quartz. Only in most outdoor work could respirators be used solely during activities that produced dust or when working near sources of dust because the outside air cleared away the dust formed quite quickly, the exception being hooded facade construction sites (Figure 6). If there was no ventilation on the premises, it could be assumed that a respirator must be worn throughout the work day in particularly dusty jobs, such as demolition hammering, drilling hollow-core slabs, grinding concrete floors, and leveling partition walls and ceilings (Figure 6). Correspondingly, if, for example, jackhammering or diamond sawing was carried out in an underpressurized section, the respirator could be removed while staying there no earlier than one hour after the jackhammering or diamond sawing ended, assuming that the air currents followed the recommendations (6 h^−1^) [29] and no other dusty work was carried out in the section, such as removal of demolition waste (see Figure 6). 

Most importantly, just as tasks potentially producing high amounts of respirable quartz dust could be carried out safely, it was also apparent that one can be exposed excessively in work tasks where dust emission is normally low by making suboptimal choices as pertains to working methods or, for instance, to the timing of the work task in relation to other work tasks with much higher dust emissions (Figure 6). Management was exposed, for example, by spending long periods of time without a respirator in departments or on floors where dusty work was done. Element installers were exposed when, for instance, performing multiple drill locations to install support bars and post-installed reinforcing bars for concrete elements without respiratory protection. The drivers of work machines could similarly be significantly exposed to respirable quartz when cabin windows are open, or by spending long periods outside the cabin when, for example, charging holes were drilled, dry crushed concrete waste was pulverized at demolition sites, or dry substrates were leveled with plate compactors, or when moving dry, dusty quartz-containing materials, such as foam glass, sand, or demolition waste. Green builders, on the other hand, could be exposed to quartz by working close to front loaders bringing dry gravel or sand, and when mixing concrete without a respirator, or when using plate compactors either to settle pavement seaming sand, or to even sand before laying pavement stones. Similarly, building cleaners were significantly exposed if they worked on premises where dusty work was done, or if they used a brush instead of a cleaning squeegee to pile up larger waste, or when emptying vacuum cleaners that did not have closable dust bags. In building cleaning, as in most other interior jobs. It was, therefore, important to consider how other jobs were paced in relation to cleaning. Further, it mattered which choices were made to control dust during the work itself. Consequently, in interior construction sites, cleaning the floor or department should be done, at the earliest, the next day after dusty work is finished (Figure 6).

For these reasons, when considering what employees to list to the ASA register, no occupation could be excluded on principle. Nor was it possible to arrange work tasks in order based on the significance of exposure. The construction site’s task-specific and work-phase-specific dust prevention plans, as well as practices and personal choices made, ultimately determined the extent to which each worker was exposed to respirable quartz (Figure 6). In terms of avoiding exposure, the most important thing was not which work task was performed but how the work was done and what dust prevention measures were followed during it. For example, in many outdoor jobs, such as the demolishing of buildings or the leveling and moving of soils, the use of water determined whether the foremen, shovelers, or landscapers were exposed to respirable quartz. Similarly, in many indoor jobs, in addition to the use of water, exposure was dependent on the ventilation rate, the use and effectiveness of equipment-specific exhausts, the timing of the work in relation to other dusty jobs, and the use of respirators and the duration of their use.

## 5. Conclusions

Most of the dust and exposure controlling measures applied on the investigated construction sites were sufficient for keeping the risks associated with quartz exposure acceptable even in tasks associated with high-quartz dust emissions. Exposure was excessive in a significant portion of the work tasks studied, with most of the high exposure being due to either not using respirators at all or not using them for long enough after the dust-emitting activity ceased. This does not mean, however, that respirators necessarily need to be used in all work tasks where excessive exposures were recorded. In some of them, other dust- or exposure-controlling measures were shown to be sufficient. Namely, for instance, the use of water to control emissions as well as using work machines with appropriately ventilated cabins in many outdoor tasks. Further, outdoors, installing flexible socks on dust separators during water drilling with drill rigs was shown to be effective as well. Continuous use of respirators could be avoided in some of the indoor tasks associated with significant or excessive exposures as well. For example, in tiling, using tile cutters instead of angle grinders and taking appropriate measures in the mixing of mortars were shown to be sufficient measures to avoid excessive exposure, provided no diamond drilling was done. Indoors, mobile air cleaners kept in the immediate vicinity of dust emission sources were an effective means to lower the respirable quartz content in the general air.

Unfortunately, however, in most of the indoor tasks where excessive exposure was found, respirators need to be used not only when actively performing dust-emitting activities but during most of the work day if mobile air cleaners, high-flow exhaust vents, or underpressurization, coupled with compartmentalization, are not used to limit the spread of dust emissions to the general air. As it stands, at least in the construction sites we investigated, most of the work tools where either water or tool-specific local exhausts, or both, were used to control dust emissions, dust leaked into the general air in significant amounts. Either as such or, initially, contained in small dust containing water droplets. Albeit, this happened in smaller amounts than if no measures were taken. Furthermore, this necessitated the use of respirators while using them. In fact, in many work tasks, respirators were needed even after the dusty activity ceased while working on the same premises.

The message our study promotes is that all construction tasks, including those where excessive exposure was found, can be safely performed while keeping the respirable quartz and dust exposures below 10% of the OELs. Provided that appropriate dust- and exposure-controlling measures are followed, there is no need for health checks—so long as training programs are in place to provide guidance in the procedures described, including the selection and use of respirators and on the condition that their implementation is monitored effectively. In all tasks where the exposure was mostly low, it was possible to be significantly exposed through the general air or by making poor choices in terms of dust control. As concerning respirable quartz exposure through the general air in such jobs, the timing of the work in relation to work tasks associated with high-dust emissions was key, as was using the right type of respirators and the timing of their use, when necessary. 

The choice of respirators was generally correct on the construction sites we investigated. However, instead, deficiencies were repeatedly found in their use and maintenance, as was also the case with the maintenance of vacuum cleaners, the planning, execution, and oversight of compartmentalization and underpressurization, as well as in the implementation of other common dust control measures, such as connecting local exhaust vents to devices. In addition, work methods with low dust emissions were not always chosen when available. For instance, a contractor might have chosen not to use water in the demolition of buildings if, according to the contract, the contractor had to pay for the water. Even so, it is our opinion that when considering overall costs to employers, including the cost of insurance premiums and occupational health care services, not to mention costs to society and the public health care system, managing risks pertaining to respirable quartz exposure in the construction industry will, in the long run, be much more economical than the realization of those risks.

## Figures and Tables

**Figure 1 ijerph-20-05431-f001:**
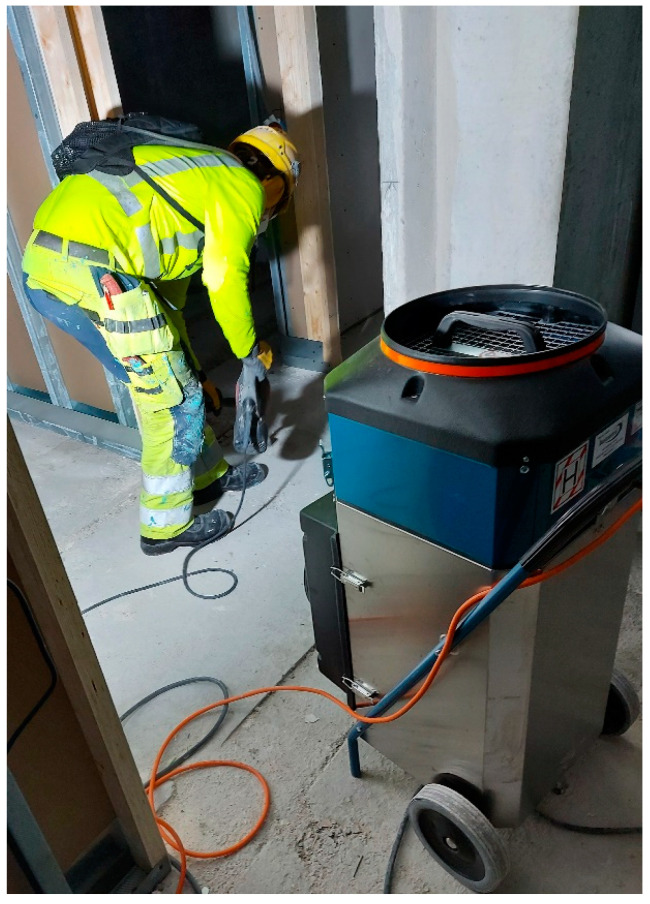
Using a mobile air cleaner to lower dust exhausts to general air from a hand-held jackhammer with a blade. Reproduced with permission from Tuomi et al., Finnish Institute of Occupational Health, 2022 [34].

**Figure 2 ijerph-20-05431-f002:**
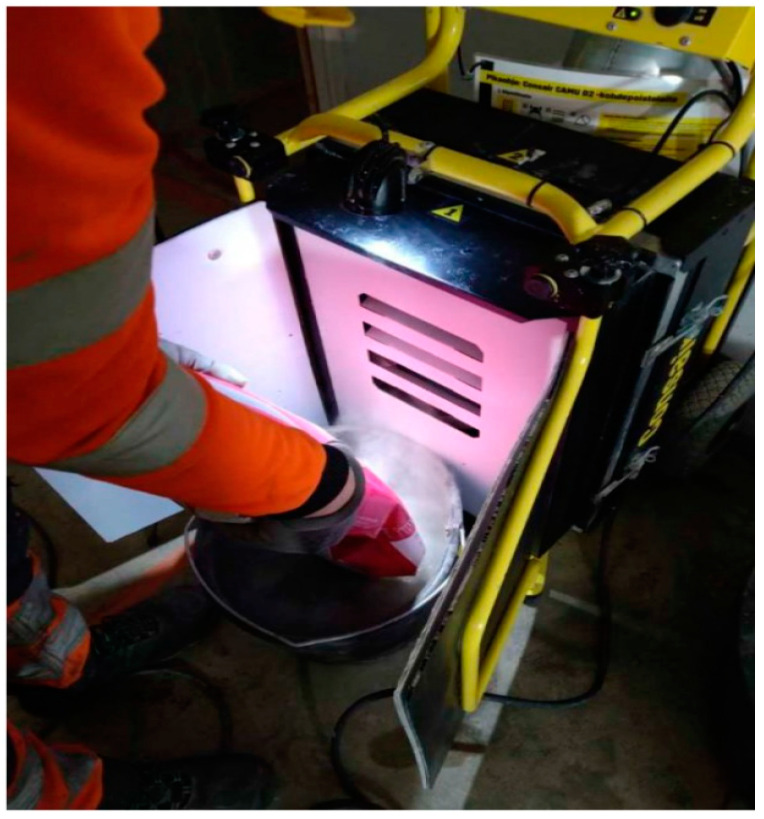
Use of a directed air cleaner in the mixing of mortar. Reproduced with permission from Tuomi et al., Finnish Institute of Occupational Health, 2022 [34].

**Figure 3 ijerph-20-05431-f003:**
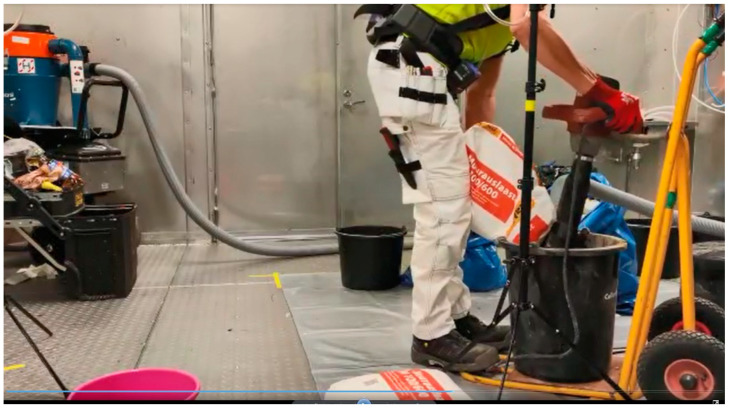
Testing an exhaust vent attached to a blender connected to an H-class vacuum cleaner. Reproduced with permission from Tuomi et al., Finnish Institute of Occupational Health, 2022 [34].

**Figure 4 ijerph-20-05431-f004:**
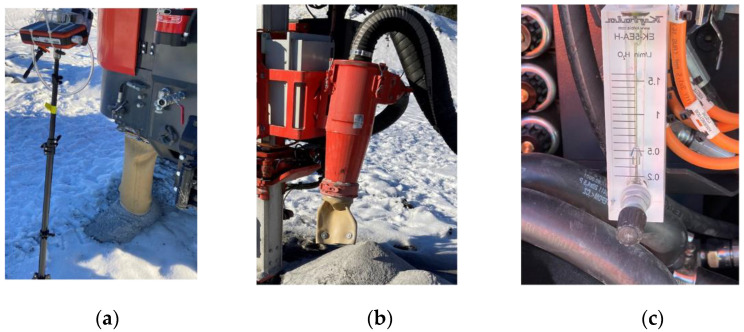
(**a**) Drill rig back separator with soc. (**b**) Front separator with soc. (**c**) Water supply adjustment of drill rig (ca. 0.4 dm^3^/min). Reproduced with permission from Tuomi et al., Finnish Institute of Occupational Health, 2022 [34].

**Figure 5 ijerph-20-05431-f005:**
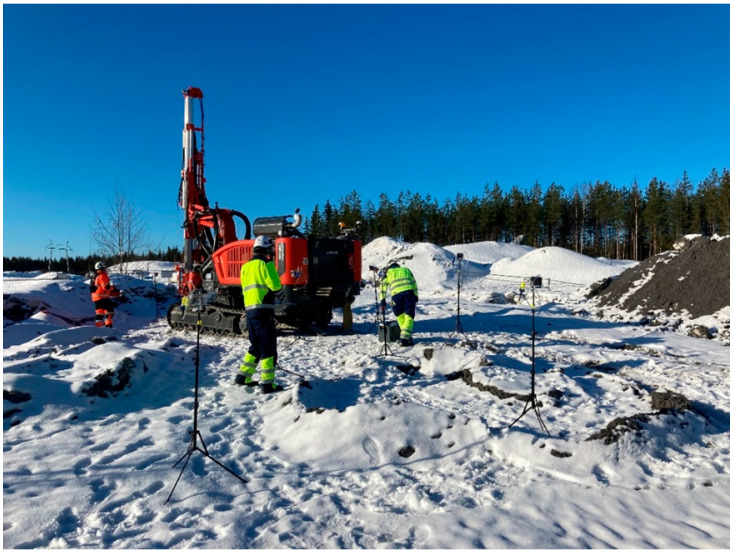
Measuring quartz dust emissions of a drilling rig in a test field. Reproduced with permission from Tuomi et al., Finnish Institute of Occupational Health, 2022 [34].

**Figure 6 ijerph-20-05431-f006:**
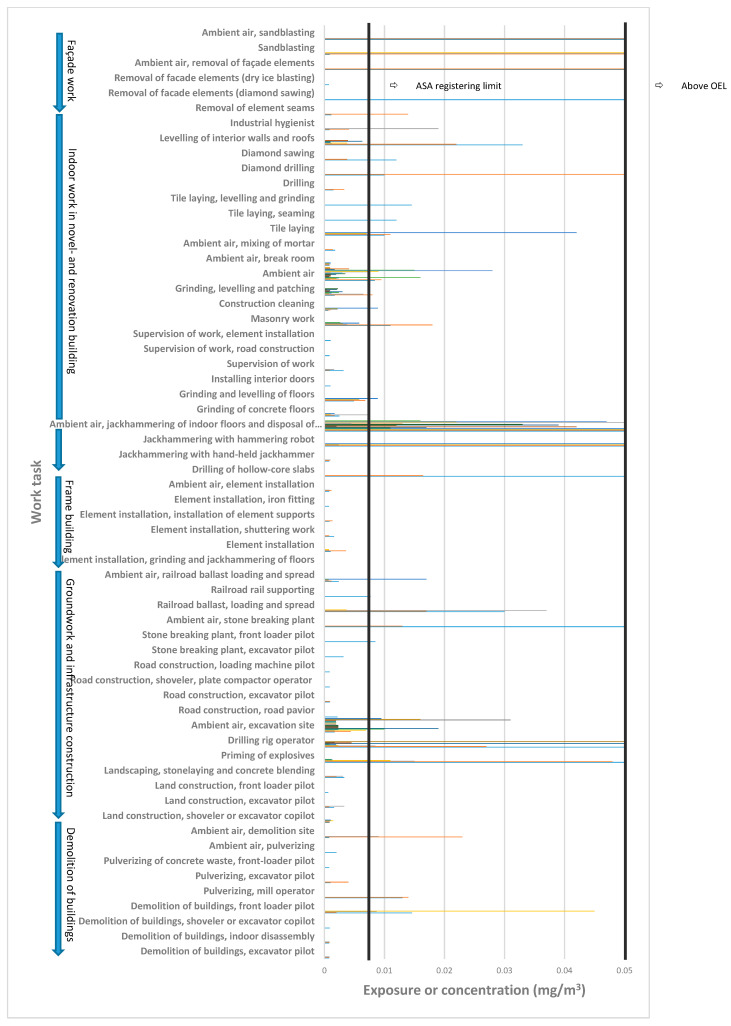
Respirable quartz exposure in work tasks, taking into account respiratory protection. Each colored line represents the estimated exposure of one worker. Reproduced with permission from Tuomi et al., Finnish Institute of Occupational Health, 2022 [34].

**Figure 7 ijerph-20-05431-f007:**
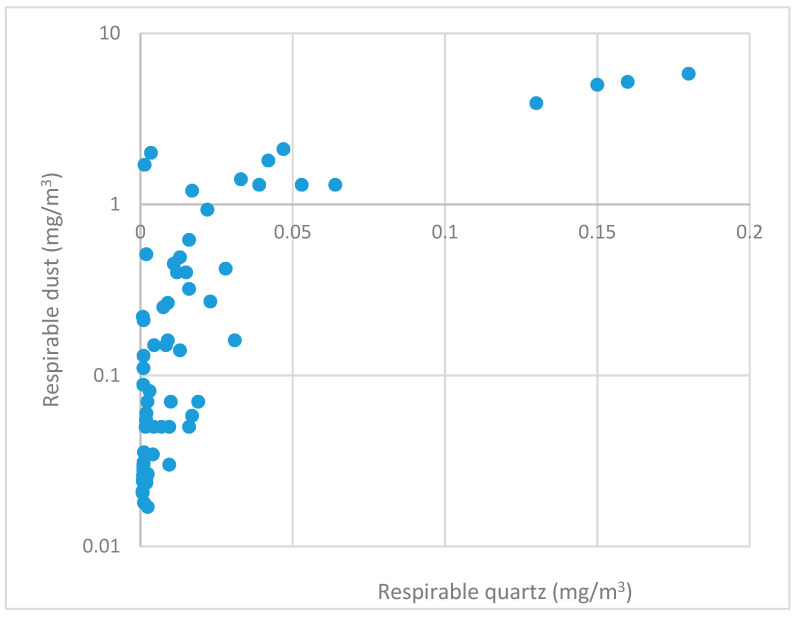
General air respirable quartz and dust concentrations at the investigated building sites. Reproduced with permission from Tuomi et al., Finnish Institute of Occupational Health, 2022 [34].

**Figure 8 ijerph-20-05431-f008:**
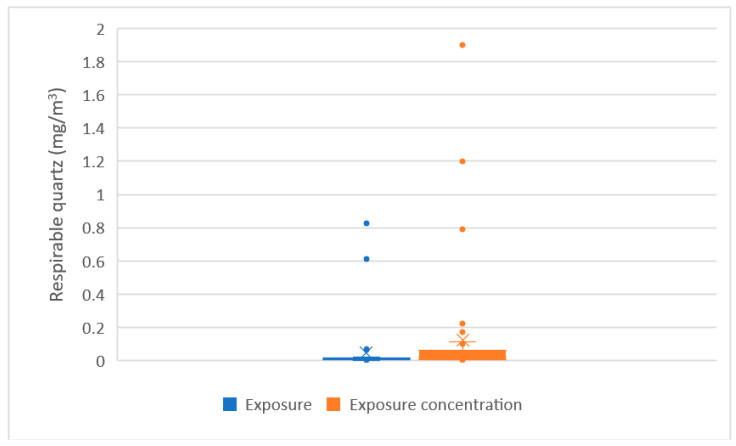
The effectiveness of using respirators during dusty activities. Reproduced with permission from Tuomi et al., Finnish Institute of Occupational Health, 2022 [34].

**Table 1 ijerph-20-05431-t001:** (**a**) Classification of quartz exposure. (**b**) Classification of exposure to respirable dust.

Exposure (mg/m^3^)	% of OEL_8h_	Significance of Exposure
**a**		
<0.005	<10%	Low
0.005–0.02	10–40%	Moderate
0.02–0.05	40–100%	Significant
0.05–0.1	>100%	Excessive
>0.1	>200%	Exceeds statutory exposure limit
**b**		
<0.1	<10%	Low
0.1–0.5	10–50%	Moderate
0.5–1	40–100%	Significant
>1	>100%	Excessive

**Table 2 ijerph-20-05431-t002:** Estimated respirable quartz and dust exposures in work tasks, taking into account respiratory protection when it was used.

	Quartz	Dust
Nr of workers	148	148
Mean (mg/m^3^)	0.032	0.64
Median (mg/m^3^)	0.0031	0.080
95% percentile (mg/m^3^)	0.072	1.8
60% percentile (mg/m^3^)	0.0040	0.12
Above 10% of OEL (%)	38	45
Exceeding OEL (%)	10	10
Exceeding statutory exposure limit (%)	4	-

**Table 3 ijerph-20-05431-t003:** Respirable quartz and dust contents of general air on construction sites.

	Quartz	Dust
Nr. of samples	88	88
Mean (mg/m^3^)	0.072	1.4
Median (mg/m^3^)	0.0024	0.07
95% percentile (mg/m^3^)	0.20	5.6
Above 10% of OEL (%)	43	45
Above 40% of OEL (%)	22	30
Exceeding OEL (%)	13	20
Exceeding statutory exposure limit (%)	10	-

**Table 4 ijerph-20-05431-t004:** Excessive exposures to respirable quartz and dust on investigated construction sites.

Phase of Construction	Work Task	Quartz Exposure (mg/m^3^)	Dust Exposure (mg/m^3^)	Reasons for Exceeding OEL
Facade work	Sandblasting, helper	0.21	5.0	No respirator
	Sandblasting	0.83	13	Respirator in use only during active blasting.
		0.073	0.073	
	Removal of facade elements (diamond sawing)	0.081	2.05	No respirator
Demolition	Excavator pilot	0.045 ^1^	1.2	High dust content in general air (cabinless excavator).
Indoor work, novel and renovation sites	Diamond drilling	0.051	0.62	No respirator. Water was used during diamond drilling but not when drilling anchor holes for diamond drill rig.
	Jackhammering of interior roofs	0.070	9.4	Respirator in use only when actively jackhammering.
		0.058	1.4	No respirator
		0.1	1.8	No respirator
		0.61	12	Respirator in use only when hammering.
	Drilling of hollow-core slabs	0.065	0.53	Respirator in use only during active drilling.
	Tiling	0.042 ^1^	0.74	No respirator during shaping of tiles with an angle grinder and drilling of runs through tiles with a diamond drill
Land and foundation work, infra-structure construction	Drilling of charging holes	0.069	0.62	No respirator. In addition, water was not used to suppress dust emissions, nor were the dust separators equipped with socks.
		0.069	0.49	
		0.17	1.1	
	Priming of charging holes	0.063	0.43	Primers worked downwind and close to the drilling rig.
		0.048 ^1^	0.27	

^1^ Upper limit of result including method uncertainty (±27%) exceeds OEL.

**Table 5 ijerph-20-05431-t005:** Respirable quartz exposure of workers in relation to respirator use.

	Respirator ^1^	No respirator ^2^
Nr. of workers	44	106
Mean	0.043	0.022
Median	1.0	1.0
95% percentile	0.073	0.069
Above 10% of OEL	50	35
Significantly exposed (% >0.02 mg/m^3^)	14	17
Exceeding OEL (%)	11	10
Exceeding statutory exposure limit (%)	5	4

^1^ Workers who used a respirator during dusty activities; ^2^ Workers who did not use a respirator.

## Data Availability

Raw data produced in the study are saved in the Laboratory Management System of the Finnish Institute of Occupational Health and can be made available upon request from the institute. The data are not publicly available due to institute policy concerning information privacy.

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
