# Peer review of "Managing Quartz Exposure in Apartment Building and Infrastructure Construction Work Tasks"

_ijerph, 2023, doi:10.3390/ijerph20085431_

Round 1

Reviewer 1 Report

General comments

The manuscript is well written, easy to read and follow, and contains enough details.

The study on which the article is based was conducted in an occupational field (construction work) and deals with a type of exposure (exposure to quartz), that are currently at the forefront of occupational health research and is of increased interest. It examines several aspects of quartz exposure in the construction industry (such as respiratory protection, emission reduction options) and provides a broad picture of the current exposure situation through the multitude of examined work processes. At the same time, it does not give a completely accurate picture in general, but rather only of the Finnish situation. Of course, this is not a weakness, but rather an asset of the study, which can be easily compared with the results of other similar studies (in other countries). In the following, I will make a few specific suggestions that can be used to improve the quality of the article. In the “Materials and Methods” part at the same time, some additional information and clarification is needed.

Specific comments

Please, consider the use of point instead of comma to indicate decimal places through the entire manuscript!

Line 28: The keywords might be reconsidered. The phrases “quartz exposure” and “construction work” are part of the title, might not be given as keywords too.

Line 218-219: It was explained later, however, it could also be noted here that two samples were taken on each worker.

Line 238-255: Please explain (e.g., with equations) how the exposures were calculated from the two measurements on each worker.

Line 248: What were the assigned protection factors? Please, give the range at least!

Line 263-293 (Sampling and analysis of samples): Please add some additional details, such as:

·         Quality and producer/supplier of the reagents used (e.g., HCl, 2-propanol, KBr)

·         How the concentration of respirable dust was measured. Details of the gravimetric analysis are required. Were there any challenges by weighing the mixed cellulose ester membrane filters? These are not the best in terms of weight stability.

Line 265: Was the diameter of the filter really 3.7 mm? I would think of 25 mm or 37 mm instead.

Line 272: It is not clear how the filtering funnel was used. What has happened with the mixed cellulose ester filter (used for the sampling) before this process? What was the material of the filter in the funnel?

Line 362-371: Here, it could be indicated what have happened with the two samples excluded from the results.

Discussion and Conclusions might be restructured. Conclusions might be shortened including only the main findings and the most important suggestions. The rest could be part of the Discussion.

Author Response

We are grateful for the comments and feel they were very helpful in improving the manuscript! Please find below a description of changes made in reference to each suggestion.

Please, consider the use of point instead of comma to indicate decimal places through the entire manuscript! We updated the manuscript as suggested, with points throughout to indicate decimal places.

Line 28: The keywords might be reconsidered. The phrases “quartz exposure” and “construction work” are part of the title, might not be given as keywords too. This is a good point and we did some adjustments accordingly.

Line 218-219: It was explained later, however, it could also be noted here that two samples were taken on each worker. We added the word “separately” (line 219) to clarify that two samples were taken, one during the workind day and one during dusty work phases. 

Line 238-255: Please explain (e.g., with equations) how the exposures were calculated from the two measurements on each worker. We did as suggested and added an equation (see lines 257-273).

Line 248: What were the assigned protection factors? Please, give the range at least! We have added the protection factors for the most frequently used respirator models (see lines 265-267)

Line 263-293 (Sampling and analysis of samples): Please add some additional details, such as: Quality and producer/supplier of the reagents used (e.g., HCl, 2-propanol, KBr); How the concentration of respirable dust was measured. Details of the gravimetric analysis are required. Were there any challenges by weighing the mixed cellulose ester membrane filters? These are not the best in terms of weight stability. Qualities and suppliers of chemicals were added as suggested. Details of gravimetric analyses were added as well together with the ISO-standard reference followed (lines 283-297). Stable conditions as suggested in ISO-standard 15767 were followed during conditioning and weighing (t 20±°C, RH 50 ± 5 %). The balance used had an internal ionization bridge. Perhaps due to these reasons, a low detection limit measured from blank filters was achieved (0,06 mg), even though mixed cellulose ester membranes were used.

Line 265: Was the diameter of the filter really 3.7 mm? I would think of 25 mm or 37 mm instead. Yes, You are of course right. 25 mm filters/samplers were used. We corrected this (line 284).

Line 272: It is not clear how the filtering funnel was used. What has happened with the mixed cellulose ester filter (used for the sampling) before this process? What was the material of the filter in the funnel? Dilute HCl, 2-propanol and deionized water were poured through the mixed cellulose ester filter containing the sampled dust using the funnel. We clarified this in the present version (lines 300-301).

Line 362-371: Here, it could be indicated what have happened with the two samples excluded from the results. No samples were excluded. The 300 samples mentioned on lines 218-219 were rounded from 296. In the present version, we corrected this (see lines 219-220).

Discussion and Conclusions might be restructured. Conclusions might be shortened including only the main findings and the most important suggestions. The rest could be part of the Discussion. We took this excellent advice and did just that. Hopefully the new version with a shorter conclusion is better.

Reviewer 2 Report

This is an important piece of work that deserves to be published. However, the text betrays its origin as a longer technical report and for that reason it is more difficult to identify the purpose of the report. Section 1.4 has an extended description of the aims, which are unclear. Is the aim to identify (jobs, workers or tasks?) that require to be reported “to the ASA register and in which tasks the statutory limit value or OEL value may be exceeded”? If so then the authors do not list these (this could be done in an online appendix). Is it to “identify work tasks where quartz exposure can be significant or excessive and to give instructions on their safe execution so that the exposure can be kept at a low level”? Also, there is a lot of material in this long paper, and I think it would be easier if the authors considered splitting the text into two papers, with the second paper perhaps concentrating on the “evaluated the effectiveness of various dust control measures to reduce exposure”.

I think the authors should be careful about how they describe the summarised their data from their investigation. As far as I can tell the sample of jobs and tasks was a priori designed to focus on the higher exposed jobs. So, they are not necessarily (probably not, see line 207) representative of Finnish construction workers in general. I know that the authors are aware of this but the reader is then led to statements like the “90% percentile of all estimated exposures was 0,05 mg/m3”, with the implication that “exposure was excessive in a significant portion of the work tasks studied” – this is not surprising since they were selected to be the higher exposures.

The description of how the authors adjusted measurements to account for the wearing of respirators is inadequate and must be improved (see Line 368). Reference is made to the use of assigned protection factors, but the authors do not seem to realise that these values are conservative, designed to indicate the minimum protection offered to a worker when wearing a respirator. The actual (average) protection will be much better than this value. Therefore, adjusting the exposure data using these data seriously overestimates the workers exposure when wearing RPE. It is also not clear to me what data have been adjusted and what data are the actual measurements, e.g. in Table 2.  The authors should reflect on what they have done and what they are attempting to achieve.

The authors treatment of measurements below the limit of detection is naïve and prone to bias, i.e. “If the limit of quantitation was not met, the result was depicted as < 2,0 μg. These results (21 %) were treated as 2,0/2 μg in the statistical calculations.” The issue of how to properly summarise such data has been extensively discussed in the occupational hygiene literature and the authors should follow a more “modern” approach, e.g. Ogden (2010).

Minor comments

Line 46 the 0.2‰ would be clearer if expressed as a percentage. Also, I think the journal style requires the decimal as a “.” Not “,”.

Line 58 “significantly rarer” what does this mean?

Line 59 “Consequently, the 95 percentile of workplace measurements” Measurements of what?

Line 265: “on 3,7 mm” – 37?

Line 268 “2,75 dm3/min.” litres are used later – why not here?

Line 270 “for a minimum of 4 h, usually close to 8 h” what was the distribution of sampling times?

Section 2.5 - Line 309: “These measurements were done at an actual indoor work site with minute to neglible ventilation.” This work in this section is poorly described. The authors might consider splitting this work into a separate paper.

Section 2.6 – this is not clear.

Line 342 – refence is made to Fig 6 – is this an erroneous refence.

Figure 6 is unhelpful, and it is not properly described in the text. What is the reader expected to take from this?

Line 377 “A significant proportion (22 %) of the general air concentrations were of magnitude associated with an increased silicosis risk (>0,2 mg/m3) when present during the span of a working career (Table 3).” The figure quoted here is different form that in the table.

Line 424 – does this statement refer to the data in Fig 7 (note, a log-scale would make it easier to interpret the data)?

Figure 8 is not particularly helpful. What is the reader meant to take from it? I see more results in the 0.005 bin for “no RPE” and then several outlier data with either “RPE” or “No RPE”.

Table 5: It’s not clear to me whether the respirator data are adjusted or not?

Fig 9: log-log scales might help

Line 459 “If there was no ventilation in the premises, it could be assumed that a respirator must be worn throughout the working day” – this message is important but seems to be contradicted later, i.e. line 526.

Line 500 “when considering what employees to list to the ASA register, no occupation could be excluded on principle”. What does this mean – all construction jobs should be reported, all “high exposed” construction jobs or any job should be reported unless there are reasons to think exposures are low? The whole of this paragraph fails to give clear guidance to employers.

Line 534 “For ex.,”

Line 540 “Unfortunately, however, for the majority of the indoor tasks where excessive exposure was a concern, respirators need to be used” What does this mean - “excessive exposure was a concern”?

Reference

Ogden TL: Handling results below the level of detection. The Annals of Occupational Hygiene 2010:1-2.

Author Response

Thank You very much for giving us excellent comments for the betterment of our manuscript!

We did restructure the discussion, shortened the conclusions and took out section 2,6 and Figure 8. However, since we as suggested added a section on how the exposure estimates were calculated (mentioning the protection factors used), the manuscript did not get much shorter. We have still opted to include all measurements in the new version of the manuscript, because we were not able to perform enough repeated intervention measurements on worksites pertaining to dust control methods to merit a separate publication. These were methods that were used in some of the worksites, but not frequently enough to yield enough data for a second publication. Also, we felt that the dust control intervention measurements performed here were needed in the present context to show that even many extremely dusty work tasks can be performed safely.

As to  whether the sample of jobs and tasks was a priori designed to focus on the higher exposed jobs, this was really not the case and we have now clarified this on lines 190, 204 and 466-474. The study did, in fact, include workers that during a typical workday are potentially exposed to quartz mainly during short and far between activities such as drilling short holes to concrete or levelling of rough patches in walls. As well as, for instance, foremen, green builders and road construction workers that for much of the working day are not exposed at all. So as stated on lines 471-474, it can be argued that the present study does give a comprehensive view of the exposure of those Finnish construction workers that can be exposed to respirable quartz at work. However, it did not include architects, planners and other workers for which exposure can basically be ruled out without performing any measurements.

As suggested, we added a description on how the exposure estimates were calculated for workers wearing respirators (lines 257-273). And with this, we  included the protection factors used. In Table two, just as in Fig 6, respiratory protection has been taken into account. We have now clarified this on lines 384-385.

As suggested, we recalculated the estimate for results below the limits of quantitation (LOQ) for the quartz and respirable dust analyses using a more modern approach (see lines 296-297 and 320-322). For quartz, the Ganser and Hewett equations/spreadsheet yielded a result of 1,2 µg instead of 1,0, which we rounded to 1 µg, to avoid recalculation of all pictures and tables. For respirable dust, the result was 0,034 mg instead of 0,030 mg which, again, was rounded to 0,03 mg. In our case, these estimates yielded basically the same results, but even if it we would have used some other method, this would not have affected the conclusions. We tested this for quartz by using different values for results below the  quantitation limit (1 µg and 1,5 µg, see table below) and the only parameter slightly affected was the median.

  Substituting  LOD (mg)
  0,001 mg* 0,0015 mg
Nr 88 88
Average 0,0722 0,0725
Median 0,0024 0,0030
95 % percentile 0,1995 0,1995
nr of results > 10 % of OEL 43 43
nr of results > OEL 13 13
> statutory limit 10 10
>40 % of OEL 22 22
*Current approach in the manuscript (from Ganser & Hewett, 2010)

Please find below answers to other specific questions as well as a description of changes made point by point.

As mentioned above, we have now elaborated how the different exposure estimates were calculated (see lines 257-273) and given examples on lines 265-267 of protection factors used. In most cases, respirators with assisted breathing having an assigned protection factor of either 200 or 1000 were used by the workers. In these cases, the respirator removed nearly all exposure during the time it was used. In the cases where a basic FFP3 mask was used, the protection factor  was 20, which means that in the calculations, quartz exposure while using the respirator was reduced to ca. 5 % of what it would have been without the respirator. Even in these cases then, most of the exposure was removed according to the estimates and hence, the exposure estimates should not be overestimates. It could rather be argued that in some individual cases the exposure estimates pertaining to particularly the use of FFP3 masks may be underestimates rather than overestimates, as we made no exceptions in the case of, for ex. bearded workers.

The authors treatment of measurements below the limit of detection is naïve and prone to bias, i.e. “If the limit of quantitation was not met, the result was depicted as < 2,0 μg. These results (21 %) were treated as 2,0/2 μg in the statistical calculations.” The issue of how to properly summarise such data has been extensively discussed in the occupational hygiene literature and the authors should follow a more “modern” approach, e.g. Ogden (2010). As mentioned above, we recalculated the limits of quantitation using the approach suggested by Ganser & Hewett (2010), but this did not affect the results or  conclusions.

Minor comments

Line 46 the 0.2‰ would be clearer if expressed as a percentage. Also, I think the journal style requires the decimal as a “.” Not “,”. As suggested, we changed commas to points throughout.

Line 58 “significantly rarer” what does this mean? This is explained in the sentence that follows it (lines 60-63): “..the 95 percentile of respirable quartz workplace exposure measurements made by the Finnish Institute of Occupational Health (FIOH) was 1.0 mg/m3 in 2006 and 0.05 mg/m3 in 2016, while the share of exposures exceeding the OEL value decreased simultaneously from 44 percent to 5 percent [4].”

Line 59 “Consequently, the 95 percentile of workplace measurements” Measurements of what? We clarified what we meant by that line (lines 60-61) “(Consequently, the 95 percentile of respirable quartz workplace exposure measurements..)”

Line 265: “on 3,7 mm” – 37? You are right of course. The correct information would have been 25 mm. We corrected this (line 284).

Line 268 “2,75 dm3/min.” litres are used later – why not here? To be consistent, we changed l/min to dm3/min throughout.

Line 270 “for a minimum of 4 h, usually close to 8 h” what was the distribution of sampling times? Usually. the measurements were started at the start of the working day ca. 7 o´clock and ended when the worker left from work around three o’clock or later. 9 out of ten times then, the measurements took about 8 h, but in a few cases the work task measured took a shorter time. For instance, an electrician might have worked at the site investigated only for half a day (ca. 4 h), in which case the sampling time was shorter. But these were exceptions.

Section 2.5 - Line 309: “These measurements were done at an actual indoor work site with minute to neglible ventilation.” This work in this section is poorly described. The authors might consider splitting this work into a separate paper. We clarified this section (lines 335-337): “These measurements were done at an actual indoor work site. As was the case in all other indoor worksites measured, there was no general ventilation in the premises”. We opted to include these measurements in this manuscript, because in the span of the present study we were not able to perform enough repeated intervention measurents on worksites to merit a separate publication. Also, these measurements were needed in the present context to show that even many extremely dusty work tasks can be performed safely.

Section 2.6 – this is not clear. This is a very good comment and we tend to agree. As this section refers to results not included here, but which were already mentioned in the introduction on lines 195-200, we decided to erase this section altogether.

Line 342 – refence is made to Fig 6 – is this an erroneous refence. As mentioned above, we erased this section (section 2.6), including the reference in question.

Figure 6 is unhelpful, and it is not properly described in the text. What is the reader expected to take from this? The Figure caption had been moved to the next page during editing. We moved it below the picture. This Figure contains all of the exposure estimates made in the study. From this picture one can see what the exposure was in all work tasks studied. Hence, we feel that the content of the manuscript would suffer greatly were we to exclude it. Nowhere else is it possible to check what the exposure ranges were in all measured work tasks. And every time we refer to the exposure being either low, moderate, significant, or excessive in a particular task, it refers to the results in this Figure. Another way to do this would have been to tabulate all results, but that would have taken much more space and would not, in our view, be as illustrative. Figure 6 also yields an overall picture of exposures in different phases of construction and enables comparison of exposures between different work tasks.

Line 377 “A significant proportion (22 %) of the general air concentrations were of magnitude associated with an increased silicosis risk (>0,2 mg/m3) when present during the span of a working career (Table 3).” The figure quoted here is different form that in the table. This is correct, which is why we added the form used in the table in the brackets for clarity (line 393): concentrations were of a magnitude associated with an increased silicosis risk (>0.2 mg/m3, >40 % of OEL).

Line 424 – does this statement refer to the data in Fig 7 (note, a log-scale would make it easier to interpret the data)? Yes it does. We did try out a logarithmic scale as well, but we felt it did not work in practice. 50 % (average) and 58 % (median) were calculated from the differences between the exposure concentrations and exposures presented in Fig. 7

Figure 8 is not particularly helpful. What is the reader meant to take from it? I see more results in the 0.005 bin for “no RPE” and then several outlier data with either “RPE” or “No RPE”. We tend to agree with this. We stated (lines 444-448) that “The distribution of quartz exposure by concentration category among workers who used respirators did not differ significantly from workers who did not use respirators (Fig. 8)..” As this can be deduced from Table 5 as well, we opted to remove Figure 8.

Table 5: It’s not clear to me whether the respirator data are adjusted or not? In this table, we compare exposure of workers who used respirators to the exposure of workers that did not use respirators. We added an explanation below the table.

Fig 9: log-log scales might help. We used a log scale here as suggested (Figure 8 in the new version).

Line 459 “If there was no ventilation in the premises, it could be assumed that a respirator must be worn throughout the working day” – this message is important but seems to be contradicted later, i.e. line 526. These statements don’t necessarily contradict one another. In the first statement (lines 507-510) we refer to particularly dusty work tasks. In the latter statement (lines 560-562), we refer to work tasks where excessive exposures were recorded. With the examples given (562-566) we aimed to explain that with appropriate dust control measures some of the work tasks that are often particularly dusty, can be made safe. But granted, when read one after the other this may seem confusing.

Line 500 “when considering what employees to list to the ASA register, no occupation could be excluded on principle”. What does this mean – all construction jobs should be reported, all “high exposed” construction jobs or any job should be reported unless there are reasons to think exposures are low? The whole of this paragraph fails to give clear guidance to employers. As seen in Figure 6, exposure in any given work task can vary greatly, depending on the dust control measures taken. In other words, whether to register a worker or not depends on the choices made to limit exposure. In the Finnish report referred to in the introduction [37], we have given exposure range estimates for each work phase measured. This was omitted from the present manuscript due to lack of space. Also, we have prepared more than thirty “good practices” for the work tasks studied and they can be found online, albeit for now, only in Finnish [38].

Line 534 “For ex.,” We changed this to “For example” (line 567).

Line 540 “Unfortunately, however, for the majority of the indoor tasks where excessive exposure was a concern, respirators need to be used” What does this mean - “excessive exposure was a concern”? We changed this to “where excessive exposure was found” (line 573-574).
